

# Quantum Chirikov criterion: Two particles in a box as a toy model for a quantum gas

**Dmitry Yampolsky[1], Nathan L. Harshman[2]\*, Vanja Dunjko[1], Zaijong Hwang[1] and Maxim Olshanii[1]**

**1** Department of Physics, University of Massachusetts Boston, Boston, MA 02125, USA
**2** Department of Physics, American University, 4400 Massachusetts Ave. NW, Washington, DC 20016, USA

⋆ harshman@american.edu

## Abstract

We consider a toy model for emergence of chaos in a quantum many-body short-range-interacting system: two one-dimensional hard-core particles in a box, with a small mass defect as a perturbation over an integrable system, the latter represented by two equal mass particles. To that system, we apply a quantum generalization of Chirikov's criterion for the onset of chaos, i.e. the criterion of overlapping resonances. There, classical non-linear resonances translate almost automatically to the quantum language. Quantum mechanics intervenes at a later stage: the resonances occupying less than one Hamiltonian eigenstate are excluded from the chaos criterion. Resonances appear as contiguous patches of low purity unperturbed eigenstates, separated by the groups of undestroyed states—the quantum analogues of the classical KAM tori.

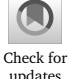

# 1 Introduction

The celebrated Chirikov resonance-overlap criterion [1, 2] constitutes a simple analytic estimate for the onset of chaos in an integrable, deterministic Hamiltonian system weakly perturbed from integrability. This criterion can be considered as a heuristic and intuitive precursor to the rigorous Kolmogorov-Arnold-Moser (KAM) theorem [3–5] that serves the same purpose for classical systems.

We consider the quantum version of the Chirikov condition because, despite varied numerical and experimental attempts to address the quantum integrability-to-chaos transition, a quantum version of the KAM theorem remains elusive. A good discussion of the challenges can be found in [6]. This reference in particular mentions the work of Hose and Taylor [7] as having proposed a criterion (albeit one dependent on both the perturbation scheme and the basis) for determining if one can assign a full set of quantum numbers to the states of a nonintegrable Hamiltonian obtained by perturbing an integrable one. A promising approach to a quantum KAM theorem is outlined in [8], but full details have never appeared. In a more recent advance, a quantum version of what are known as Nekhoroshev estimates was constructed in [9].

Although the full quantum KAM theorem is still unavailable, the Chirikov criterion has yielded to quantum formulation. In [10–13], the authors study a quantum system with exactly two nonlinear resonances, while in [14, 15], the studied quantum system is truncated to two resonances. Physically, both systems consist of a single particle driven by an external field. The classical counterparts of both systems feature a chaotization threshold.

Here we apply the quantum Chirikov criterion to a simple two-dimensional system: two one-dimensional hard-core particles in a box, with a small mass defect. We consider all nonlinear resonances that are present, with no truncations. Note that, unlike in the previous studies, our system is self-contained and isolated, with no external fields. The mass defect is treated as a perturbation over the integrable system with two equal-mass particles. In [16] it has been found that in a many-body case, relaxation in a one-dimensional two-mass mixture occurs in a few collisions per particle, similarly to a multidimensional gases of hard-core spheres. On the other hand, by analogy with other few-body toy models [17], this two-body model allows us to estimate the relaxation threshold in a many-body setting [9].

Traditionally, research on KAM theory avoids considering short-range interactions because they typically lead to no chaos threshold, c.f. [18]. Indeed, as we show below, the classical counterpart of our model is chaotic for any nonzero perturbation strength. However, in the quantum model the threshold for the emergence of chaos is restored even for a system with short-range interactions.

# 2 Chirikov condition

For two-dimensional classical systems, the Chirikov criterion [1, 2] can be formulated as follows. Let

$$H(\theta_1, I_1, \theta_2, I_2) = H_0(I_1, I_2) + \epsilon V(\theta_1, I_1, \theta_2, I_2),$$

be the Hamiltonian of an integrable system $H_0$ weakly perturbed by a correction $\epsilon V$, $2\pi$-periodic with respect to both $\theta_1$ and $\theta_2$. Here $I_{1,2}$ and $\theta_{1,2}$ are the corresponding canonical actions and angles, respectively. (Note that upon quantization, the actions $I_{1,2} = \hbar n_{1,2}$ become the two quantum numbers for the eigenstates of the quantum version of $H_0$.) Consider a resonance point $(\bar{I}_1, \bar{I}_2)$ in the action space where the frequencies $\omega_{1,2} \equiv \partial H_0/\partial I_{1,2}\big|_{\bar{I}_1, \bar{I}_2}$ of

the two subsystems are in a rational relationship:

$$\frac{\omega_2}{\omega_1} = \frac{p}{q},\tag{1}$$

where $p$ and $q$ are assumed to be mutually prime. In the resonant approximation, the non-resonant terms in the double Fourier decomposition of the perturbation can be replaced by a constant leading to

$$V(\theta_1, I_1, \theta_2, I_2) \approx V_{p,q}(p\theta_1 - q\theta_2, I_1, I_2) + \text{const.}$$

The Hamiltonian now depends on a single function of the two coordinates, indicating integrability. Indeed, under the canonical transformation

$$\begin{aligned}
\theta_1 &= (p^2 + q^2)^{-1}(p\theta + q\tilde{\theta}), \\
I_1 &= \bar{I}_1 + p\mathcal{I} + q\tilde{I}, \\
\theta_2 &= (p^2 + q^2)^{-1}(-q\theta + p\tilde{\theta}), \\
I_2 &= \bar{I}_2 - q\mathcal{I} + p\tilde{I},
\end{aligned}$$

the action $\tilde{I}$ becomes the sought-after second integral of motion.

For a sufficiently weak perturbation, assume the motion is bound to a narrow region in the action space. Accordingly, the resonant Hamiltonian emerges when the Taylor expansion of the terms $H_0$ and $V$ are truncated to the second and the zeroth order in $\mathcal{I}$, respectively, while $\tilde{I}$ is kept at zero. The Hamiltonian becomes $H \approx \mathcal{H} + \text{const.}$, where the *resonant Hamiltonian* $\mathcal{H}$ for the $(p,q)$ resonance at $\bar{I}_1, \bar{I}_2$ has the form

$$\mathcal{H} = \frac{\mathcal{I}^2}{2\mathcal{J}} + \epsilon\mathcal{V}(\theta),\tag{2}$$

with $1/\mathcal{J} = \partial^2 H_0/\partial\mathcal{I}^2\big|_{\bar{I}_1, \bar{I}_2}$ and $\mathcal{V}(\theta) = V_{p,q}(p\theta_1 - q\theta_2, \bar{I}_1, \bar{I}_2)$.

In the resonant Hamiltonian (2), the linear term in $\mathcal{I}$ term is absent because $\mathcal{I}$ controls shifts in the action space that are *tangential* to the equienergy surface. As a result, the time evolution generated by the Hamiltonian (2) describes a motion *along* the surface. A bounded-motion trajectory, with energies $\mathcal{E}$ in the range $\mathcal{E} \in [\min_\theta \mathcal{V}, \max_\theta \mathcal{V}]$ (see Fig. 1), occupies a finite-width segment of the equienergy surface and is called a *nonlinear resonance*. These bound trajectories differ qualitatively from their unperturbed counterparts, and they signify an appearance of the destroyed tori in the KAM theory. The unbounded trajectories that live outside of the $\mathcal{E} \in [\min_\theta \mathcal{V}, \max_\theta \mathcal{V}]$ energy region remain close to their unperturbed versions.

According to the Chirikov theory, each point $(I_1, I_2)$ on the equienergy surface of interest must be tested for belonging to a resonance. If it does not belong to any, no matter what $p$ and $q$ are, this point constitutes a mobility-limiting, impenetrable boundary on the equienergy surface, analogous to a KAM torus. To the contrary, if every point in a particular region of the equienergy surface belongs to *at least two* resonances, the whole region is conjectured to be chaotic.

Quantization of the Chirikov criterion [10–13, 15] follows a scheme analogous to the classical case. The perturbation is truncated to resonant matrix elements only. Furthermore a semiclassical limit is assumed, where it becomes possible to reformulate the problem in terms of an effective quantum Hamiltonian,

$$\hat{\mathcal{H}} = \frac{(\hat{\mathcal{I}} + \hbar\delta)^2}{2\mathcal{J}} + \epsilon\mathcal{V}.\tag{3}$$

The action $\mathcal{I}$ takes the form of an angular momentum-like operator $\hat{\mathcal{I}} \equiv -i\hbar\frac{\partial}{\partial\theta}$ and is quantized as $\mathcal{I} = \hbar m$, $m = 0, \pm1, \pm2, \ldots$. The additional increment $\hbar\delta$ is a possible quantum offset

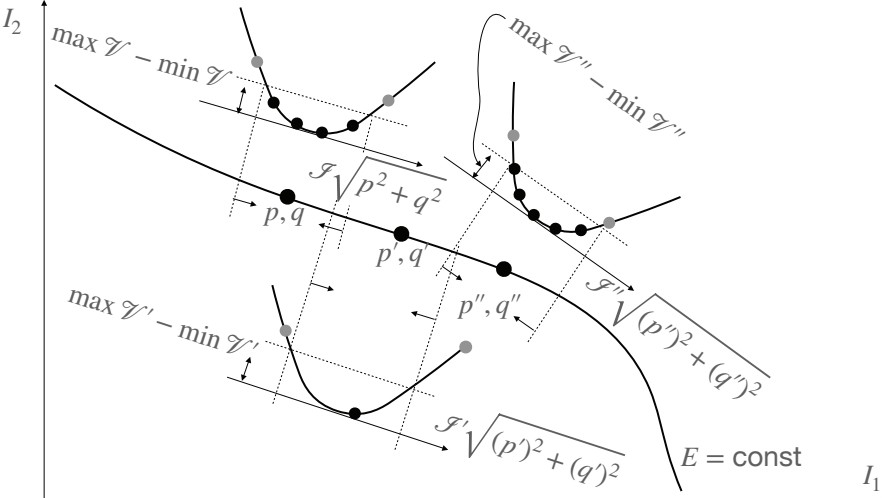

Figure 1: A schematic view of the Chirikov criterion for the onset of chaos (see text). For three resonances $(p,q)$, $(p',q')$, $(p'',q'')$ on an equienergy surface, the effective resonant Hamiltonian for bound states and the non-linear resonances are depicted. The parabolas signify the unperturbed energy along a line tangential to the equienergy surface with energy $E$ at the resonance point. The horizontal coordinate, $\mathcal{I}\sqrt{p^2+q^2}$ (see (2) for the meaning of the resonance action $\mathcal{I}$), is chosen in such a way that the distance between two points on the tangential line is equal to the distance between their counterparts on the $(n_1, n_2)$ plane. For a classical system, these three resonances would constitute a fragment of a chaotic region by the Chirikov criterion because the non-linear resonances overlap. In the quantum version of the system, the two resonances $(p,q)$ and $(p'',q'')$ contain more than one unperturbed energy eigenstate that survive quantization (represented by black dots). However, they are separated by a $(p',q')$ resonance that contains one or fewer unperturbed eigenstates, meaning that this resonance disappears under quantization. Therefore, in contrast to the classical system, for the quantum system the resonances $(p,q)$ and $(p'',q'')$ are separated by a "KAM torus" given by the unpopulated resonance $(p',q')$ and as such remain isolated.

from the mismatch of the quantized unperturbed states and the resonant equienergy surface. The overlap of the resonances is studied in a manner identical to the classical case.

In our article, we deliberately chosen a system that is classically chaotic at all energies (see Appendix A.3 and references [18, 19]). The system is two one-dimensional particles in a box whose masses are almost identical. The mass defect plays a role of a perturbation of the integrable system of two identical particles. Our main interest is the question of whether quantization affects the above conclusion. What we found was that not all the resonances identified in the classical system are eligible to being included in the resonance overlap consideration. While classically, a resonance can occupy an arbitrary small region of the phase space, quantum-mechanically this volume is limited to $(2\pi\hbar)^d$, where $d$ is the number of spatial dimensions. Classical resonances that occupy less than this volume become indistinct from the unperturbed eigenstates: they will be discarded from the Chrikov criterion of the overlapping resonances (see Fig. 1). As a result, a perturbation strength threshold for quantum chaos emerges, $\epsilon \sim \bar{n}^{-2/3}$, absent in the classical case. Here, $\epsilon$ is a dimensionless relative mass defect, and $\bar{n}$ is a typical quantum number of one degree of freedom.

# 3  Quantum two-particle model

For two one-dimensional hard-core particles with slightly different masses in a box, the quantum version of the Chirikov condition exactly follows this minimal extension of the classical condition. We see the onset of chaos, as observed by the onset of Wigner-Dyson statistics for the energy level, occur where the mass difference (considered as a perturbation) crosses the threshold given by the quantum Chirokov condition and and the quantum analogs of the classical KAM tori are broken; c.f. Fig. (2). See the Appendix A.3 for a Chirikov analysis of a classical analogue of our system.

Consider the two one-dimensional hard-core particles of masses and $M_2 > M_1$, with coordinates $x_1 \leq x_2$, moving in a hard-wall box of size $L$. This system is often recast as a right triangular billiard (see, e.g., [19]), where after a change of variables, a two-dimensional scalar-mass particle emerges, moving in a triangle with angles $\pi/2$, $\alpha/2$, $\pi/2 - \alpha/2$ with $\alpha/2 = \arctan\left[\sqrt{M_2/M_1}\right]$. However to frame the mass difference as a perturbation, the Hamiltonian

$$\hat{H} = \hat{H}_0 + \epsilon \hat{V}, \tag{4a}$$

can be expressed in the form

$$\hat{H}_0 = \frac{1}{2M_0}(\hat{p}_1^2 + \hat{p}_2^2), \tag{4b}$$

$$\hat{V} = -\frac{1}{2M_0}(\hat{p}_2^2 - \hat{p}_1^2), \tag{4c}$$

where $\hat{p}_{1,2} \equiv -i\hbar\partial/\partial x_{1,2}$ are the particle momenta, $1/M_0 = 1/(2M_1) + 1/(2M_2)$, and $\epsilon = (M_2 - M_1)/(M_1 + M_2)$. The familiar spectrum of the unperturbed Hamiltonian (4b) has eigenenergies

$$E^{(0)}_{(n_1,n_2)} = T_0(n_1^2 + n_2^2),$$

and eigenstates

$$\Psi^{(0)}_{(n_1,n_2)}(x_1, x_2) = \phi_{n_1}(x_1)\phi_{n_2}(x_2) - \phi_{n_2}(x_1)\phi_{n_1}(x_2),$$

with $\phi_n(x) = \sqrt{2/L}\sin(k_n x)$, $k_n = \pi n/L$, and $1 \leq n_1 < n_2$. The energy scale $T_0$ is given by $T_0 \equiv \hbar^2\pi^2/(2M_0 L^2)$.

The energy-scaled matrix elements of the perturbation (4c) in the unperturbed basis $|(n_1, n_2)\rangle$,

$$v_{(n_1,n_2),(n_1',n_2')} \equiv \frac{1}{T_0}\langle(n_1,n_2)|\hat{V}|(n_1',n_2')\rangle, \tag{5}$$

are zero unless the sum $n_1 + n_2 + n_1' + n_2'$ is odd. When the sum is odd, we find the expression

$$
\begin{aligned}
v_{(n_1,n_2),(n_1',n_2')} = \ & \frac{256}{\pi^2}\Big[n_1 n_1' n_2 n_2'(n_1^2 - n_2^2)\big((n_1')^2 - (n_2')^2\big)\Big]\Big/\Big[(n_1 + n_1' + n_2 + n_2') \\
& \times (n_1 + n_1' - n_2 - n_2')(n_1 - n_1' + n_2 - n_2')(n_1 - n_1' - n_2 + n_2') \\
& \times (n_1 + n_1' + n_2 - n_2')(n_1 + n_1' - n_2 + n_2') \\
& \times (n_1 - n_1' + n_2 + n_2')(-n_1 + n_1' + n_2 + n_2')\Big],
\end{aligned}
$$

which simplifies to the approximate result

$$v_{(n_1,n_2),(n_1',n_2')} \approx \frac{4}{\pi^2}\frac{N_1^2 - N_2^2}{\Delta n_1^2 - \Delta n_2^2}, \tag{6}$$

when $\Delta n_{1,2} \ll N_{1,2}$ with $N_{1,2} \equiv (n_{1,2} + n'_{1,2})/2$ and $\Delta n_{1,2} \equiv n_{1,2} - n'_{1,2}$. Note that while $N_{1,2}$ can be both integer and half-integer, the numbers $\Delta n_{1,2}$ are strictly integer.

The perturbation (4c) breaks the integrability of $H_0$ and the new eigenstates $\Psi_\lambda$ of the full Hamiltonian (4a) obeying $\hat{H}\Psi_\lambda = E_\lambda \Psi_\lambda$ can be decomposed into sums over the unperturbed eigenstates $\Psi^{(0)}_{(n_1,n_2)}$ using the expansion coefficients $\langle \lambda|(n_1,n_2)\rangle$ as

$$\Psi_\lambda(x_1, x_2) = \sum_{(n_1, n_2)} \langle \lambda|(n_1, n_2)\rangle \Psi^{(0)}_{(n_1, n_2)}(x_1, x_2).$$

For a sufficiently strong perturbation strength $\epsilon$, there are eigenstates $\Psi_\lambda$ of the perturbed Hamiltonian (4a) that consist of broad superpositions of the unperturbed eigenstates $\Psi^{(0)}_{(n_1,n_2)}$. Our goal is to interpret such states in terms of the nonlinear resonances of the Chirikov theory [1, 2]. Similarly to the classical case, we will attempt to interpret an overlap between the resonances as an onset of chaos.

Let us construct the quantum analogy to a classical nonlinear resonance of order $p:q$ by constructing lines of unperturbed states tangent to the equienergy surface. Two mutually prime integers $p$ and $q$ define a ray in $(n_1, n_2)$ space pointing out from the origin (Fig. 6(a)). Each unperturbed state $(n_1, n_2)$ lies on a 'resonance line' with slope $-q/p$ perpendicular to the $p:q$ ray. The point of intersection of the resonance line through $(n_1, n_2)$ with the $p:q$ ray occurs at a position $(\bar{n}_1, \bar{n}_2) = \big(qk/(p^2+q^2), pk/(p^2+q^2)\big)$, where the positive integer $k = qn_1 + pn_2$ serves as a convenient label for the resonance line through $(n_1, n_2)$. Denote the energy at the intersection point $(\bar{n}_1, \bar{n}_2)$ by $\bar{E} = T_0 k^2/(p^2+q^2) \equiv T_0 \bar{n}^2$, with $\bar{n} \equiv \sqrt{\bar{n}_1^2 + \bar{n}_2^2}$.

Generally, a resonance line intersects multiple unperturbed states. If $(n_1, n_2)$ is on resonance line $k$, then so is $(n_1+pj, n_2-qj)$ for all integers $j$ such that the constraints $0 < n_1 < n_2$ are fulfilled. Denote the unperturbed state on resonance line $k$ that is closest to the $p:q$ ray by $(n_1^*, n_2^*) = (\bar{n}_1 + \delta p, \bar{n}_2 - \delta q)$ and the other states on resonance line $k$ by $(n_1^* + mp, n_2^* - mq)$. The increment $\delta$ can be interpreted as the quantum offset from the classical resonance point to the lowest unperturbed quantum state on the same resonance tangent line (see Fig. 1). Note that by this construction, each unperturbed state is uniquely identified by a pair $(k, m)$

$$(n_1, n_2)_{(k,m)} = (\bar{n}_1 + (m+\delta)p, \bar{n}_2 - (m+\delta)q).\tag{7}$$

The energy of the unperturbed eigenstates on the $k$ resonance line of the $p:q$ resonance now reads

$$E_m = \bar{E} + T_0(p^2 + q^2)(m+\delta)^2.\tag{8}$$

For $p, q \sim 1$, the prefactor of the parabola is as *small* as the ground state energy. In contrast to states on the same resonance line $k$, the typical energy distance between neighboring unperturbed energy states is $\bar{n}$ times greater, and thus, a even a relatively small perturbation can potentially couple a range of $m$ indices in the vicinity of $m = 0$. The resulting eigenstates of the perturbed system will then be represented by large, multicomponent superpositions of the unperturbed states. Such broad eigenstates constitute quantum "nonlinear" resonances, the quantum analogues of the classical nonlinear resonances. Indeed at the point $(\bar{n}_1, \bar{n}_2)$, the classical frequencies $\omega_{1,2} \equiv \partial E^{(0)}_{(n_1,n_2)}/\partial(\hbar n_{1,2})$ obey the classical resonance condition (1)[1].

In what follows, we truncate the Hilbert space to only the states lying on a particular resonance line $k$. Then, let us interpret the index $m$ in (8) as a momentum index of a fictitious one dimensional particle on a ring of a circumference $2\pi$. The Hamiltonian for such a fictitious particle coincides with the conjectured expression (3), where the moment of inertia $\mathcal{J}$ is given

---

[1]Such a resonance is qualitatively different from the "quantum resonance" proposed in [20]. An example of that kind a resonance in our system would be $T_0/(\hbar\omega_1) = p/q$. While our condition (1) is, so far, completely classical, the "quantum resonance" of [20] has no classical analogue.

by $\hbar^2/(2\mathcal{J}) = T_0(p^2+q^2)$ and the quantum offset $\delta$ is defined above. The potential energy $\epsilon\mathcal{V}$ in (3) can be inferred from the matrix elements (5)-(6) restricted to states on the resonance line $k$ with different $m$:

$$\mathcal{V}_{m,m'} \equiv \langle (n_1, n_2)_{(k,m)} | \hat{V} | (n_1, n_2)_{(k,m')} \rangle. \tag{9}$$

Our eventual goal is to find the width of the resonance $m_{\max}$ that in analogy to the classical construction [1,2] corresponds to the momentum width of the separatrix trajectory, the boundary between the bound and unbound motion. As a first step, we consider the approximation (6) valid for $\Delta n_{1,2} \ll N_{1,2}$ and require that the states with $|m| \lesssim m_{\max}$ yield this inequality. Substituting the definition of the quantum numbers $(k, m)$ for unperturbed states on the same resonance line (7) into the inequality, and assuming $p \sim q$ and $m \sim m'$, we obtain the following condition[2]

$$m_{\max} \ll \frac{\bar{n}}{\sqrt{p^2+q^2}} = \frac{k}{(p^2+q^2)}.$$

This condition for the validity of the approximation (6) has to be verified a posteriori for each $p{:}q$ resonance line $k$.

When the condition $|m|, |m'| \ll \bar{n}/\sqrt{p^2+q^2}$ is met, then the perturbation matrix elements (9) simplify to

$$\mathcal{V}_{m,m'} = \begin{cases} 0 & \text{for} \quad m-m' = \text{even} \\ -\frac{4}{\pi^2} T_0 \frac{\bar{n}^2}{p^2+q^2} \frac{1}{(m-m')^2} & \text{for} \quad m-m' = \text{odd} \end{cases}.$$

The above are precisely the matrix elements of a potential

$$\mathcal{V}(\theta) \overset{\epsilon \ll 1}{\approx} -\mathcal{V}_0(1 - 2|\theta|/\pi), \qquad -\pi \le \theta < +\pi \tag{10}$$

between the unperturbed eigenstates $\Phi_m(\theta) = \frac{1}{\sqrt{2\pi}} \exp[im\theta]$ with $\mathcal{V}_0 = T_0\bar{n}^2/(p^2+q^2)$ (see Fig. 5). Interestingly, the Hamiltonian (3) is identical to its classical counterpart obtained using a resonant approximation (see Appendix A.2).

Physically, a quantum "nonlinear" resonance would manifest itself in appearance of bound states of (3)—similarly to the classical case (2). Energetically, only the states for which the kinetic energy $\hat{\mathcal{I}}^2/(2\mathcal{J})$ does not exceed the span of the potential, $|\mathcal{V}(\theta = \pm\pi) - \mathcal{V}(\theta = 0)|$ can participate in such bound states. More specifically, we require that span of the unperturbed kinetic energy defined in (8), $T_0(p^2+q^2)m_{\max}^2$, be equal the span of the potential energy (10), $2T_0\bar{n}^2/(p^2+q^2)$. This condition limits the state index $m$ to $|m| \lesssim m_{\max}$, with

$$m_{\max} = \sqrt{\epsilon}\,\frac{\sqrt{2}}{p^2+q^2}\bar{n}. \tag{11}$$

Notice that for a sufficiently small perturbation parameter $\epsilon$, the necessary condition $m_{\max} \ll \bar{n}/\sqrt{p^2+q^2}$ for establishing (10) will be automatically satisfied for all $\epsilon$ that yield $\epsilon \ll \sqrt{p^2+q^2}$. Recall that by construction, $\epsilon$ can not be greater than unity, and the $\sqrt{p^2+q^2}$ border can only be reached for a combination of extreme mass ratios and small $p$ and $q$.

So far, the quantum "nonlinear" resonances described above were constituting a one-to-one copy of the classical phenomenon [1,2]. The quantum limitation emerges from a requirement for the resonance to occupy more than one unperturbed eigenstate:

$$m_{\max} \gtrsim (m_{\max})_{\min} \sim 1. \tag{12}$$

---

[2]See Appendix B for a special case of 1:0 resonance.

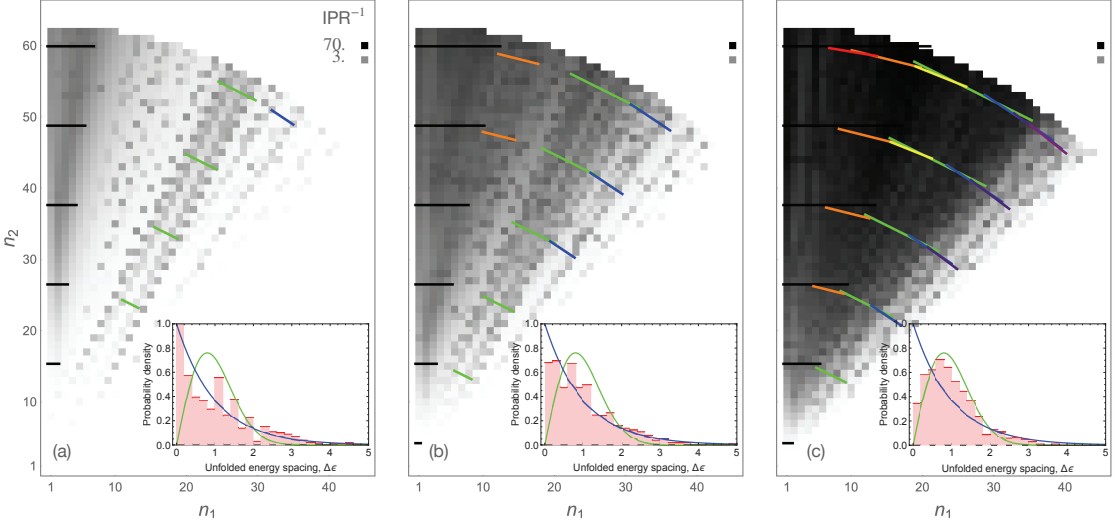

Figure 2: The inverse purity, $\text{IPR}^{-1} \equiv \left(\sum_{\lambda} |\langle\lambda|n_1, n_2\rangle|^4\right)^{-1}$ (i.e. the inverse of the inverse participation ratio IPR), of the eigenstates $|(n_1, n_2)\rangle$ of two one-dimensional equal mass hard-core particles in a box with respect to the eigenstates $|\lambda\rangle$ of the same system but with a small mass defect, $\epsilon \equiv (M_2 - M_1)/(M_2 + M_1)$. The values of $\epsilon$ are 0.006 (a), 0.02 (b), and 0.06 (c). Darker squares represent the unperturbed states destroyed by the perturbation, while the lighter one—the analogues of the classical KAM tori—are insensitive to it. Colored lines show the location of the classical resonances, further post-selected under a requirement that a resonance must span more than one unperturbed quantum state. Quantum post-selected resonances are depicted for the following $p : q$ ratios: $1 : 0$ (black), $2 : 1$ (green), $3 : 2$ (blue), $4 : 1$ (orange), $4 : 3$ (indigo), $5 : 2$ (yellow), $6 : 1$ (red), and $5 : 4$ (purple). The smaller the sum $p^2 + q^2$, the earlier a resonance $p : q$ appears. The quantum lower bound on the resonance width has been chosen to be $(m_{\text{max}})_{\text{min}} = 0.5$. The insert shows level statistics for a range of perturbed energies $E = T_0 \bar{n}^2$ with $37.1 < \bar{n} < 52.0$, in comparison with the Poisson (green) and Wigner-Dyson (blue) distributions.

Trivial as it is, such a limitation dramatically depletes the set of allowed resonances. According to (11)-(12), in order to have *any* resonances on an equienergy surface of radius $\bar{n}$, on the $n_1 - n_2$ plane, one needs to have

$$\epsilon \gtrsim \epsilon_{\text{first resonance}} \sim \frac{1}{\bar{n}^2}.$$

To the contrary, the classical analogue of our system yields the Chirikov criterion for chaos for all $\epsilon > 0$ (see Appendix A.3).

The condition for the resonances to overlap everywhere—with no gaps—and hence, according to the Chrikov criterion, for chaos to occur is even more stringent. In the Appendix A.3, we invoke the density of the classical resonances and then, at the final stage, apply the quantum limitation (11)-(12). This analysis leads to the following estimate for the chaos threshold:

$$\epsilon \gtrsim \epsilon_{\text{no gaps}} \sim \frac{1}{\bar{n}^{\frac{2}{3}}} \sim \frac{\hbar^{\frac{2}{3}}}{M_0^{\frac{1}{3}} \bar{E}^{\frac{1}{3}} L^{\frac{2}{3}}}. \tag{13}$$

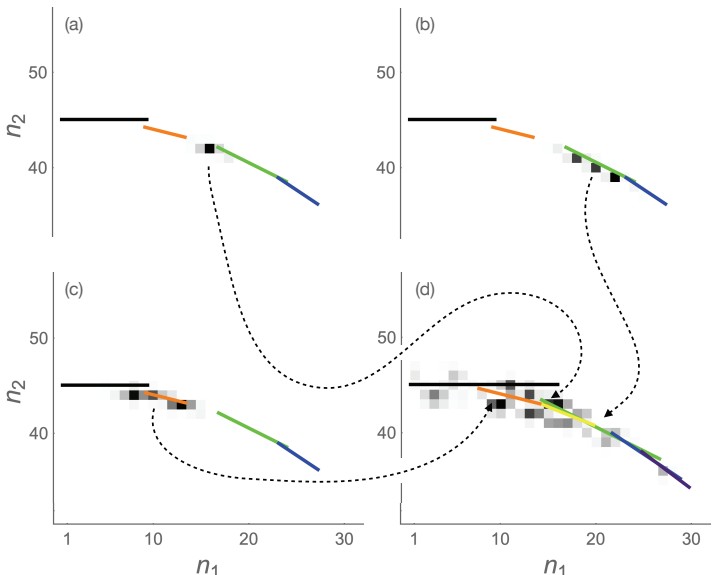

Figure 3: An example of two separate quantum nonlinear resonances, $2:1$ (b) and $4:1$ (c), separated by a KAM gap (a), at $\epsilon = 0.02$, overlap and fuse into a single, broad eigenstate (d) that also comprises the already existing $1:0$ and $3:2$ resonances and newly emerged $4:3$, $5:2$, $6:1$, and $5:4$ ones, at $\epsilon = 0.06$ (see caption of Fig. 2 for the color scheme and for the value of $(m_{\max})_{\min}$). Grey scale reflects contributions $|\langle \lambda | (n_1, n_2) \rangle|^2$ of the individual unperturbed states $|(n_1, n_2)\rangle$ to the perturbed state $|\lambda\rangle$. For clarity, on each of the four plots, the grey scale spans the whole range between white and black, white corresponding to zero overlap and black corresponding to the maximal value of the overlap, $\max_{n_1, n_2} |\langle \lambda | (n_1, n_2) \rangle|^2 = 0.614, 0.315, 0.194,$ and $0.070$ for (a), (b), (c), and (d) respectively.

Recall that above, $\bar{n} \equiv \sqrt{n_1^2 + n_2^2}$ is typical state index, $M_0$ is the unperturbed particle mass, $\bar{E}$ is the system energy of interest, and $L$ is the size of the box to which our two-body system is confined.

As expected, minimal perturbation strength (13) for chaos to occur tends to zero if $\hbar$ is moved to zero. Note that this conclusion seemingly contradicts an observation in [19] that indicates that generic right-triangular billiards partially retain memory of the initial conditions. More precisely, for any initial velocity orientations, there will be several hundred uniformly distributed orientations the trajectory visits with an anomalously high probability. We conjecture that Chirikov's criterion employed in our paper is too imprecise an instrument to detect such a subtle effect.

The above expression for the chaos threshold can be further improved (see Appendix A.3), leading to

$$\epsilon \gtrsim \epsilon_{\text{no gaps}} \approx \frac{\pi^{\frac{8}{3}}}{32} \frac{((m_{\max})_{\min})^{\frac{2}{3}}}{\bar{n}^{\frac{2}{3}}}, \tag{14}$$

where, again $(m_{\max})_{\min} \sim 1$, and its precise value is a matter of convention.

Fig. 2(b) demonstrates that when $\epsilon$ crosses the 'no gaps' threshold, the quantum-post-selected classical nonlinear resonances begin to overlap and the the level statistics undergoes

a transition from the Poisson (Fig. 2(a)) to the Wigner-Dyson (Fig. 2(c)) type, signifying an integrability-to-chaos transition. The quantum analogues of the classical resonances appear as contiguous patches of low purity unperturbed eigenstates. These results were obtained using exact diagonalization, within a 4950-state strong basis of eigenstates of an integrable reference system, represented by two equal mass balls in a box (see [21, 22] for an overview of the method).

Note that while the Wigner-Dyson statistics of Fig. 2(c) is reached at a perturbation strength $\epsilon = 0.06$, the Chirikov prediction for the chaos threshold, Eq. (14), computed for $(m_{\text{max}})_{\text{min}} = 0.5$ (the same as at Figs. 2-3), at $\bar{n} = 44.5$ (i.e. in the middle of the band used for the level statistics) gives $\epsilon = 0.7$.

An alternate perspective is depicted in Fig. 3, which shows that classical nonlinear resonances can constitute a meaningful taxonomy of the *perturbed* eigenstates in the intermediate between integrability and chaos regime. Interestingly, even when two or more resonances fuse together at stronger perturbation (Fig. 3(d)), the constituent resonances allow one to predict the location and the width of the resulting state. Intriguingly, the appearance of the eigenstates that span significant portions of the available state space (Fig. 3(d)) can also be seen as an approach towards the Eigenstate Thermalization [23–25] for the observables $\hat{p}_1$ and $\hat{p}_2$ (see (4a)) and functions thereof. Indeed, classically a thermal state would feature a uniform distribution of velocity orientations. Eigenstate Themalization would then predict that every eigenstate is spread uniformly along the equienergy surface.

## 4 Conclusions and outlook

In summary, for a Hamiltonian system, we constructed quantum analogues of the classical nonlinear resonances. The set of quantum "nonlinear" resonances must be post-selected to exclude those resonance lines containing only one quantum eigenstate or none at all. Such post-selection leads to a chaos suppression at low energies. This suppression stands in contrast to Anderson localization [26] which would manifest itself in appearance of multi-state resonances that are shorter than classically predicted; we observed no evidence for this effect in our system. We extend the notion of the Chirikov criterion for the onset of chaos to the quantum case and show that resonance overlap remains a sensitive predictor for the onset of chaos even in the quantum case. The classical resonances appear as bands of low purity unperturbed quantum eigenstates separated by the undestroyed ones—the quantum analogues of the KAM tori. We hope that our work can be used to advance understanding of a KAM threshold in low-dimensional cold quantum gases [9].

In particular, following an analogy with a toy model [17], one may attempt to estimate a quantum thermalization threshold in a one-dimensional two-mass mixture of hard-core particles [16]. We proceed as follows. We consider (i) the threshold (14), (ii) estimate $\hbar^2 \bar{n}^2 / M_0 L^2$ as a single-particle energy, (iii) associate the latter with the temperature $k_{\text{B}} T$, and (iv) regard $1/L$ as an estimate for the one-dimensional density $n_{\text{1D}}$, to obtain (v) the following threshold:

$$\epsilon \gtrsim \left( \frac{\hbar^2 n_{\text{1D}}^2}{k_{\text{B}} M T} \right)^{\frac{1}{3}}, \tag{15}$$

where $M$ is the particle mass scale, $\epsilon \sim \Delta M / M$ is a dimensionless mass defect, and $k_{\text{B}}$ is the Boltzman constant.

## Acknowledgements

The authors thank Artem Volosniev for useful discussions on numerical methods, Bala Sundaram and Alexey Tonyushkin for fruitful discussions on quantum chaos, and Svetlana Jitomirskaya for discussions on constrained distributions of mutually primes.

**Funding information**   This work was supported by the NSF (Grants No. PHY-1912542, and No. PHY-1607221) and the Binational (U.S.-Israel) Science Foundation (Grant No. 2015616).

## A    Classical analysis of two-particle model

### A.1    Set-up

Consider two one-dimensional hard-core particles of masses $\tilde{m}_1$ and $\tilde{m}_2$, with coordinates $x_1 < x_2$, moving in a hard-wall box of size $L$ (Fig. 4 (a)). The Hamiltonian for the system has the form

$$H = H_0 + \epsilon V \,, \tag{16}$$

with

$$H_0 = \eta^{(0)}(p_1^2 + p_2^2) \,, \tag{17}$$
$$V = -\eta^{(0)}(p_2^2 - p_1^2) \,, \tag{18}$$

and

$$0 \leq x_1 \leq x_2 \leq L \,,$$

where $p_{1,2}$ are the particle momenta, $1/M_0 = 1/(2M_1) + 1/(2M_2)$, $\epsilon = (M_2 - M_1)/(M_1 + M_2)$ and $\eta^{(0)} \equiv 1/(2M_0)$. In what follows, we will assume $\epsilon \ll 1$ and treat $V$ as a perturbation.

One can perform a canonical transformation of the phase-space coordinates so that in the new coordinates, the unperturbed Hamiltonian $H_0$ describes free motion in an infinite space. This transformation is performed in two steps.

1. At the first stage, we unfold the unperturbed motion in the variables of the configuration space triangle $0 \leq x_1 \leq x_2 \leq L$ to that inside the square $-L \leq \rho_1, \rho_2 \leq L$; see Fig. 4(b). This is accomplished by the following canonical (thus invertible) transformation of the phase-space coordinates:

$$\vec{r} = \hat{g}_{\vec{\rho}} \cdot \vec{\rho} \,,$$
$$\vec{p} = (\hat{g}_{\vec{\rho}})^{-1} \cdot \vec{\pi} \,,$$

with $\vec{\rho} \equiv (\rho_1, \rho_2)$, $\vec{r} \equiv (x_1, x_2)$, $\vec{\pi} \equiv (\pi_1, \pi_2)$, and $\vec{p} \equiv (p_1, p_2)$. The linear transformation $\hat{g}_{\vec{\rho}}$ is one of the eight elements of the point symmetry group of the square that brings $\vec{\rho}$ to the chamber $0 < x_1 < x_2$. The conjugate transformation $(\hat{g}_{\vec{\rho}})^{-1}$ restricts the new canonical momenta so that they now reside in the domain $0 < \pi_1 < \pi_2 < \infty$. Under this transformation, the Hamiltonian becomes

$$H = H_0 + \epsilon V \,,$$

with

$$H_0 = \eta^{(0)}(\pi_1^2 + \pi_2^2) \,,$$
$$V = -\eta^{(0)}(\pi_2^2 - \pi_1^2)\,\text{sign}(|\rho_2| - |\rho_1|) \,. \tag{19}$$

2. The second transformation unfolds the square $-L \leq \rho_1, \rho_2 \leq L$ to an infinite two-dimensional space; see Fig. 4(c). This one-to-many, non-invertible transformation emerges when one attempts to solve the unperturbed evolution of the $\rho_1, \rho_2$ coordinates using the method of images:

$$\rho_1 = \mathfrak{r}_1 \bmod_{-L} 2L \,,$$
$$\rho_2 = \mathfrak{r}_2 \bmod_{-L} 2L \,,$$
$$\vec{\pi} = \vec{\mathfrak{p}} \,,$$

with $\vec{\mathfrak{p}} \equiv (\mathfrak{p}_1, \mathfrak{p}_2)$. Here and below, $a \bmod_d b \equiv a - b \lfloor \frac{a-d}{b} \rfloor$ is the modulo function with an offset, and $\lfloor \ldots \rfloor$ is the floor function. The domain of the new momenta $0 < \mathfrak{p}_1 < \mathfrak{p}_2 < \infty$ is unchanged, but now the positions $\vec{\mathfrak{r}} \equiv (\mathfrak{r}_1, \mathfrak{r}_2)$ range over the whole plane $\vec{\mathfrak{r}} \in \mathbb{R}^2$. Under this transformation, the Hamiltonian becomes

$$H = H_0 + \epsilon V \,,$$

with

$$H_0 = \eta^{(0)} (\mathfrak{p}_1^2 + \mathfrak{p}_2^2) \,,$$
$$V = -\eta^{(0)} (\mathfrak{p}_2^2 - \mathfrak{p}_1^2) \operatorname{sign}(|\mathfrak{r}_2 \bmod_{-L} 2L| - |\mathfrak{r}_1 \bmod_{-L} 2L|) \,. \tag{20}$$

## A.2 Nonlinear resonances

In these new coordinates, nonlinear resonances are identified using temporal averaging of the perturbation $V$ over the unperturbed motion, i.e. a constant velocity propagation along a straight line parallel to a particular momentum vector $\vec{\mathfrak{p}}$. To calculate this time average, consider an unperturbed trajectory with momentum vector $(\tilde{\mathfrak{p}}_2, \tilde{\mathfrak{p}}_1)$ that crosses a point $\mathfrak{r}_1' = \mathfrak{r}_1'^{(0)}, \mathfrak{r}_2' = 0$ (see Fig. 4(c) for notations). The temporal average of the perturbation (20) becomes

$$\overline{V} = -\eta^{(0)} (\tilde{\mathfrak{p}}_2^2 - \tilde{\mathfrak{p}}_1^2) \overline{\operatorname{sign}(|\mathfrak{r}_2 \bmod_{-L} 2L| - |\mathfrak{r}_1 \bmod_{-L} 2L|)} \,.$$

The temporal average of the sign-function is related to the probability $\operatorname{Prob}_{\mathrm{grey}}(\tilde{\mathfrak{p}}_1, \tilde{\mathfrak{p}}_2, \mathfrak{r}_1'^{(0)})$, defined as follows: Consider a straight line parallel to $(\tilde{\mathfrak{p}}_1, \tilde{\mathfrak{p}}_2)$ that crosses a point $\mathfrak{r}_1' = \mathfrak{r}_1'^{(0)}, \mathfrak{r}_2' = 0$. Choose a point on this line at random. The probability of interest becomes

$$\operatorname{Prob}_{\mathrm{grey}}(\tilde{\mathfrak{p}}_1, \tilde{\mathfrak{p}}_2, \mathfrak{r}_1'^{(0)}) \equiv \text{Probability of landing on a gray square, Fig. 4(c)} \,.$$

This probability gives the temporal average through the relation

$$\overline{\operatorname{sign}(|\mathfrak{r}_2 \bmod_{-L} 2L| - |\mathfrak{r}_1 \bmod_{-L} 2L|)} = 2 \operatorname{Prob}_{\mathrm{grey}}(\tilde{\mathfrak{p}}_1, \tilde{\mathfrak{p}}_2, \mathfrak{r}_1'^{(0)}) - 1 \,.$$

Below, we list the relevant results, omitting the derivation:

(i) When the ratio of momentum components

$$\frac{\tilde{\mathfrak{p}}_2}{\tilde{\mathfrak{p}}_1} = \frac{p}{q} \,, \tag{21}$$

is given by $p$ and $q$, a mutually prime integers of *opposite parity*. Recall that the unperturbed trajectory given by (21) corresponds to a resonance between the two degrees of freedom in the system, see Eq. (1). For this case, the probability depends on the particular value of the intercept $\mathfrak{r}_1'^{(0)}$:

(a) We find

$$\text{Prob}_{\text{grey}} = \frac{1}{2} + \frac{1}{2(p^2 - q^2)},$$

when

$$\mathfrak{r}_1^{\prime(0)} = 2l(p, q) \times \text{integer},$$

where

$$l(p, q) \equiv \frac{p + q}{\sqrt{p^2 + q^2}} L.$$

(b) We find

$$\text{Prob}_{\text{grey}} = \frac{1}{2} - \frac{1}{2(p^2 - q^2)},$$

when

$$\mathfrak{r}_1^{\prime(0)} = 2l(p, q) \times \left(\text{integer} + \frac{1}{2}\right).$$

(c) For the remaining values of $\mathfrak{r}_1^{\prime(0)}$, the probability $\text{Prob}_{\text{grey}}$ is given a linear interpolation between the cases (a) and (b).

Overall, the averaged perturbation assumes the form

$$\overline{V} = -\frac{\bar{E}}{p^2 + q^2} \, \text{saw}\left[\frac{\mathfrak{r}_1^{\prime(0)}}{l(p, q)}\right], \tag{22}$$

where $\text{saw}[\xi]$ is function with period 2 and in the interval $-1 < \xi < +1$ takes the form $\text{saw}[\xi] = 1 - 2|\xi|$. The reference energy $\bar{E}$ will defined later. Remark that after a trivial coordinate transformation $\mathfrak{r}_1^{\prime(0)}/l(p, q) = \theta/\pi$, the potential $\overline{V}(\theta)$ becomes identical to the form (10) inferred from the quantum version of the problem.

(ii) In all other cases,

$$\text{Prob}_{\text{grey}} = \frac{1}{2},$$

rendering a vanishing averaged perturbation:

$$\overline{V} = 0.$$

Consider a particular pair of mutually prime opposite parity integers, $p$ and $q$, and rotate the coordinates in such a way that the "$y$ axis" coincides with the direction governed by (21):

$$\begin{aligned}
\mathfrak{r}_1' &= \vec{\mathfrak{r}} \cdot \vec{e}_1' \\
\mathfrak{r}_2' &= \vec{\mathfrak{r}} \cdot \vec{e}_2', \\
\mathfrak{p}_1' &= \vec{\mathfrak{p}} \cdot \vec{e}_1', \\
\mathfrak{p}_2' &= \vec{\mathfrak{p}} \cdot \vec{e}_2',
\end{aligned}$$

with

$$\begin{aligned}
\vec{e}_1' &\equiv \frac{1}{\sqrt{p^2 + q^2}} \begin{pmatrix} +p \\ -q \end{pmatrix} \\
\vec{e}_2' &\equiv \frac{1}{\sqrt{p^2 + q^2}} \begin{pmatrix} +q \\ +p \end{pmatrix}.
\end{aligned}$$

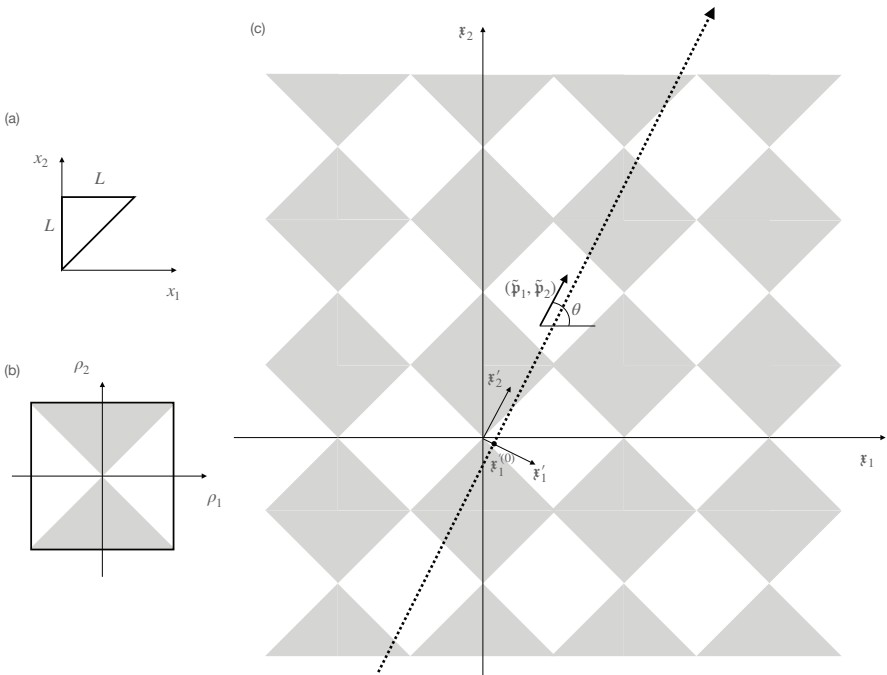



Figure 4: Three coordinate systems used to find and analyze classical nonlinear resonances. (a) The original system of coordinates. The size of the box is $L$. Particle 2 is assumed to be to the right from particle 1. (b) Unfolding the triangle to a square. The original triangle is unfolded, using mirror reflections about the $x_1 = 0$ cathetus and the $x_1 = x_2$ hypothenuse, to a square of side $2L$. While the coordinate space is enlarged by a factor of 8, momenta are constrained to a $0 < \pi_1 < \pi_2$ sector. There are no reflective walls inside the square; the outer wall, generated by the $x_2 = L$ cathetus, remains. Grey areas correspond to positive values of the function $\text{sign}(|\rho_2| - |\rho_1|)$; this function is a part of the perturbation (19). (c) Unfolding the square to a plane. The square of subfigure (b) is unfolded to a plane, via sequential mirror reflections with respect to its walls. Reflective walls disappear completely. The grey areas correspond to the positive values of the function $\text{sign}(|\mathfrak{x}_2 \bmod_{-L} 2L| - |\mathfrak{x}_1 \bmod_{-L} 2L|)$; this function is a part of the perturbation (20).

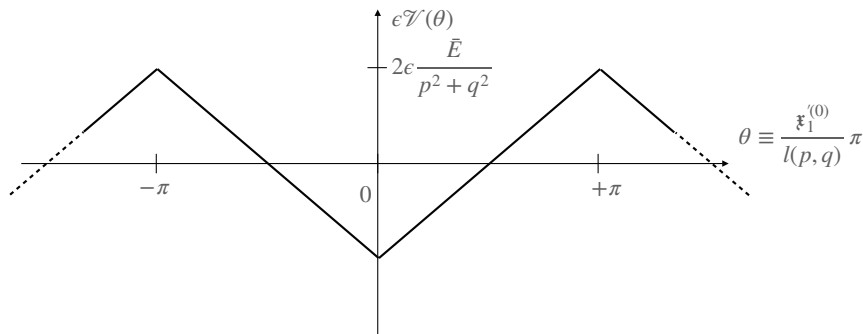

Figure 5: The form of the potential that appears in the study of a nonlinear resonance in our system.

Now, for a sufficiently small perturbation parameter $\epsilon$, motion along the $\vec{e}_2'$ axis can be approximated by its unperturbed counterpart:

$$\mathfrak{p}_2' \approx \bar{p} = \text{const},$$

$$\mathfrak{r}_2' \approx \frac{\bar{p}}{M_0}\, t\,.$$

Furthermore, the perturbation $V$ can be replaced by its time-averaged value,

$$V \approx \overline{V}\,.$$

The Hamiltonian becomes

$$H \approx \bar{E} + \mathcal{H}(\mathfrak{r}_1', \mathfrak{p}_1')\,,$$

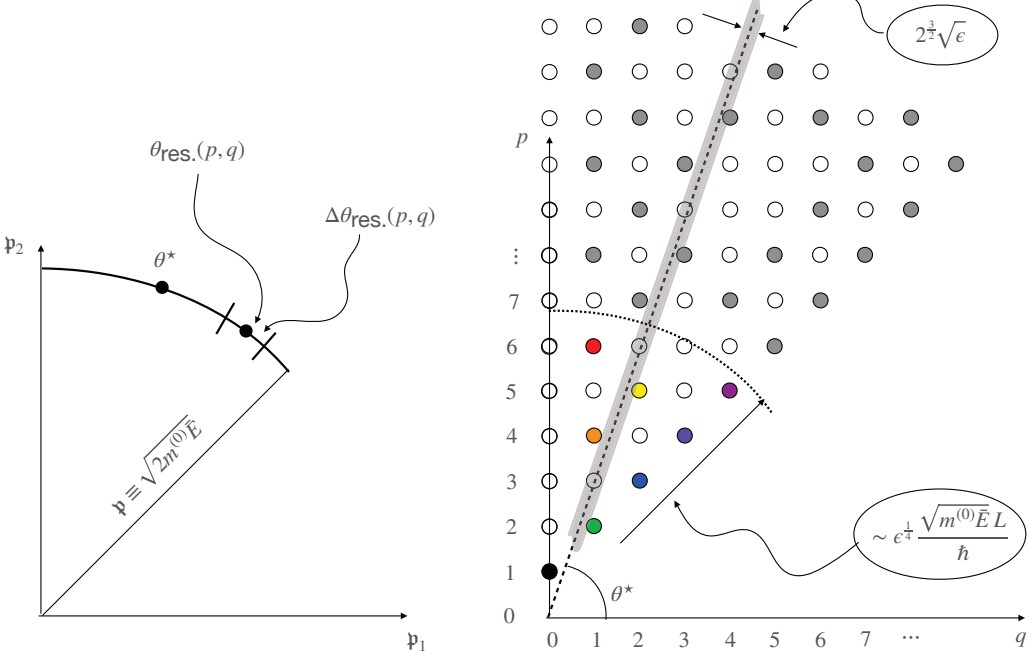

Figure 6: Steps in construction of the classical and quantum Chirikov criteria for an offset of chaos. (a) A point of on an equienergy surface $\bar{E}$ in the momentum space, at an angle $\theta^\star$ to the horizontal, is shown to lie *outside* of a nonlinear resonance $p:q$. (b) As shown in the text, the angular width of a resonance $p:q$, on an equienergy surface, is proportional to $1/\sqrt{p^2 + q^2}$. As a result, if a point $\theta^\star$ belongs to a resonance $p:q$, the point $(q, p)$ must lie in a *stripe*, in the $q, p$ space. The total width of the stripe can be shown to be $2^{\frac{3}{2}}\sqrt{\epsilon}$. Classically, the stripe is infinitely long, and any point $\theta^\star$ belongs to an infinite number of allowed resonances (colored and grey points, color scheme is the same as at Figs. 2-3. Every point on the equienergy surface turns out to be dynamically connected to any other point there resulting in chaos. Quantum mechanics sets an upper bound on the length of the stripe: $\sqrt{p^2 + q^2} \lesssim \epsilon^{1/4}\bar{n}$, with $\bar{n}$ being the typical quantum number along any of thee two directions. As a result, both the appearance of the first resonance (at $\epsilon \gtrsim \sim 1/\bar{n}^2$) and the chaos threshold (at $\epsilon \gtrsim 1/\bar{n}^{\frac{2}{3}}$) requires a *finite strength perturbation*.

where

$$\bar{E} \equiv \frac{\bar{p}^2}{2M_0},$$

and

$$\mathcal{H}(\mathfrak{r}_1', \mathfrak{p}_1') \equiv \frac{\mathfrak{p}_1'^2}{2m^{(0)}} + \epsilon \overline{V}(\mathfrak{r}_1'), \tag{23}$$

with $\overline{V}(\mathfrak{r}_1')$ given by (22). Finally, applying a canonical transformation

$$\theta = \frac{\mathfrak{r}_1'}{l(p, q)}\pi,$$
$$\mathcal{I} = \frac{\mathfrak{p}_1' l(p, q)}{\pi},$$

we arrive at the classical analogue of the quantum resonant Hamiltonian (3), with the quantum offset $\delta$ being neglected. Recall that the momentum $\mathcal{I}$ is the classical analogue of the quantum number $m$, i.e. $\hbar m \to \mathcal{I}$ (see Fig. 5).

Let us return to the form (23). Notice that the behavior of our system will crucially depend on the magnitude of the momentum $\mathfrak{p}_1'$. For

$$|\mathfrak{p}_1'| > 2\sqrt{\epsilon}\frac{\sqrt{M_0 \bar{E}}}{\sqrt{p^2 + q^2}},$$

the motion along $\mathfrak{r}_1'$ is unbounded, remaining close the unperturbed scenario. For

$$|\mathfrak{p}_1'| < 2\sqrt{\epsilon}\frac{\sqrt{M_0 \bar{E}}}{\sqrt{p^2 + q^2}},$$

however, the one observes oscillation about the resonant ratio $p/q$ between the momentum components: the motion along the $\mathfrak{r}_1$ and $\mathfrak{r}_2$ become phase-locked. All of the above is completely analogous to the setting for a nonlinear resonance described in Chirikov's original paper [1].

## A.3 Chirikov criterion

The idea of Chirikov criterion for emergence of chaos [1] is as follows. An individual resonance constitutes a strong but yet regular perturbation of a trajectory of the underlying integrable system. At a formal level, each resonance can be studied individually, via a truncation of the Fourier spectrum of the perturbation to resonant terms only. Such a treatment may or may not be self-consistent. Each resonance occupies a segment of the phase space. If every point of the segments belongs only to the resonance in question, this resonance is regarded to be an isolated one, and no chaos is expected. However, it it so happens that a given point belongs simultaneously to two or more resonances, an appearance of a mobility is inferred. When every point of the energetically allowed shell in the phase space is found to belong to two or more resonances, a global chaos is predicted at this energy.

From the geometry depicted in Fig. 6, one deduces that the angular half-width of a resonance in an equienergy surface is

$$\Delta\theta_{\text{res.}}(p, q) \approx \sqrt{2}\frac{\sqrt{\epsilon}}{\sqrt{p^2 + q^2}}.$$

To see this, consider a point at the equienergy surface $\bar{E}$, at an angle $\theta^\star$ to the horizontal. For a resonance $p : q$ to contain the point, it needs to lie—in the $q, p$ space—in an interval of a (full) length

$$2\Delta\theta_{\text{res.}}(p, q)\sqrt{p^2 + q^2} = \Delta r \equiv 2^{\frac{3}{2}}\sqrt{\epsilon}\,,$$

around a point $\theta^\star$, on a surface of a constant $\sqrt{p^2 + q^2}$. In general any allowed resonance in a stripe of a width $2^{\frac{3}{2}}\sqrt{\epsilon}$, along the $\theta^\star$ ray, in the $q, p$ space, contains the probe point $\theta^\star$ in the momentum space.

Classically, Chirikov criterion predicts that in our system, motion is always chaotic, no matter how small $\epsilon$ is. Indeed, the density of the allowed resonances where $q$ and $p$ are opposite parity and mutually prime is $(6/\pi^2) \times (2/3)$. The first factor is the probability for two randomly chosen integers to be mutually prime [27] and the second selects the opposite parity mutually prime pairs. Quantum Mechanics—through the condition (12) that resonance occupies at least one quantum state—limits the "radius" $\sqrt{p^2 + q^2}$ by

$$\sqrt{p^2 + q^2} < r_{\text{quant.}} = 2^{\frac{1}{4}}\epsilon^{\frac{1}{4}}\frac{\sqrt{\bar{n}}}{\sqrt{(m_{\text{max}})_{\text{min}}}}\,.$$

A threshold for the onset of chaos requires that any point $\theta^\star$ on the energy surface of interest belongs to at least one resonance. Thus

$$\Delta r\ r_{\text{quant.}} \approx 1\,,$$

leading to the chaos condition (13). However, for a given energy $\bar{E}$, the limit $r_{\text{quant.}}$ tends to infinity in the classical limit $\hbar \to 0$. Hence, classically the system is predicted, according to the Chirikov criterion, to be always chaotic, *no matter how small the perturbation is*.

## B  Special case of the $1:0$ resonance

In the case of $p = 1$, $q = 0$, the resonance is located near the point $(n_1, n_2) = (0, \bar{n})$. This point requires a different approximation for the matrix elements of the perturbation. As before, the resonant Hamiltonian reads

$$\hat{\mathcal{H}} = \hat{\mathcal{T}} + \epsilon\mathcal{V}(\theta)\,, \tag{24}$$

with

$$\hat{\mathcal{T}} = \frac{\hat{\mathcal{I}}^2}{\mathcal{J}}\,, \tag{25}$$

where the angular momentum $\mathcal{I}$ is defined as $\mathcal{I} \equiv -i\hbar\partial/\partial\theta$, the moment of inertia $\mathcal{J}$ is given by $1/(2\mathcal{J}) = T_0(p^2 + q^2)$. Notice that for this resonance, the $\delta$ correction vanishes and the unperturbed state is exactly on the resonance line.

An important change however is that now the "angular momentum" $\mathcal{I}$ is strictly positive,

$$\mathcal{I} > 0\,,$$

and it can no longer interpreted as $\mathcal{I} \equiv -i\hbar\partial/\partial\theta$. This follows from the resonance line (7) with $\bar{n}_1 = 0$ and $p = 1$, that gives

$$m = 1, 2, 3, \ldots.$$

A different approximation will be required to evaluate the matrix elements of the perturbation. Assume that the yet unknown resonance width $m_{\text{max}}$ obeys

$$m_{\text{max}} \ll \bar{n}\,. \tag{26}$$

In the case $p = 1$, $q = 0$, this condition, to be verified a posteriori, allows to simplify the perturbation matrix elements (Eqns. 5 and 6 in main text) as:

$$
\begin{aligned}
\mathcal{V}_{m,m'} &\equiv \langle m, \bar{n} | \hat{V} | m', \bar{n} \rangle \\
&\stackrel{m, m' \ll \bar{n}}{\approx} -\frac{4}{\pi^2} T_0 \frac{\bar{n}^2}{p^2 + q^2} \left\{ \begin{array}{lll} 0 & \text{for} & m - m' = \text{even} \\ 1 & \text{for} & m - m' = \text{odd} \end{array} \right\} \left\{ \frac{1}{(m - m')^2} - \frac{1}{(m + m')^2} \right\},
\end{aligned} \tag{27}
$$

(see Eq. 7 for $p = 1$, $q = 0$, and $\bar{n}_1 = 0$).

The following can be shown however. Let us keep the potential described by (10), but redefine the basis states as $\Phi_m(\theta) = (1/\sqrt{\pi}) \sin[m\theta]$. Physically, the Hilbert space a resonance spans is now associated with the odd function subspace of the Hilbert space of periodic functions of $\theta$; the Hamiltonian acting on this space remains exactly the same. After some algebra, one arrives at the matrix elements (27). The expression for the resonance width (11) remains thus unaltered.

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
