# Peer review of "Quantum Chirikov criterion: Two particles in a box as a toy model for a quantum gas"

_SciPost Physics, doi:SciPost Phys. 12, 035 (2022)_

## Round 3 · Referee Report · Anonymous (Referee 3) · 2021-11-18

Report

The authors have substantially improved the manuscript, and considered my suggestions seriously (some of which coincided with another reviewer's comments). I am happy to recommend this work for publication in SciPost Physics.

---

## Round 3 · Referee Report · Anonymous (Referee 1) · 2021-11-20

Report

I am happy to recommend this article for publication in SciPost Physics. The authors responded meaningfully to my criticisms and made substantial alterations to the article that, in my opinion, enhance its readability and reproducibility.

---

## Round 3 · Referee Report · Anonymous (Referee 2) · 2021-11-28

Report

The authors sufficiently answered all of my questions and remarks. They also provided corresponding extensions to the manuscript. In my opinion, the current version of the manuscript is suitable for publication in SciPost Physics.

---

## Round 3 · Author Response

Dear editors,

We thank the referees for their thoughtful comments and have made revisions along the lines of their suggestions. For the sake of thoroughness, we have included the referee comments below and added our responses where appropriate. See below.

These changes have improved the article and we thank the editors for this opportunity.

For the authors,

Nate Harshman, Vanja Dunjko, and Maxim Olshanii

---

## Round 3 · List of Changes

Referee 1

The article ``Quantum Chirikov criterion: Two particles in a box as a toy model for a quantum gas'' by Dmitry Yampolsky, N. L. Harshman, Vanja Dunjko, Zaijong Hwang, and Maxim Olshanii concerns the emergence of quantum chaos in a 1D system of two particles in a box subject to a mass perturbation, as characterised by a quantum extension of the Chirikov criterion. I found the manuscript interesting and believe that it contains results worthy of publication, but found several issues with the manuscript's clarity. Before I can recommend this article for publication, I would appreciate if the authors could address the following points.

  1. It is not clear from the text which aspects of the quantum Chirikov criterion are original to this work and which are derived from previous work. On line 33, the authors cite references [9, 11-14] without a thorough discussion of prior work regarding quantum Chirikov criteria. Without further clarification, it is unclear from the text whether the novelty of this work stems entirely from the application of a quantum Chirikov criterion to the particular system, or if there is a substantial original contribution to the development of a quantum Chirikov criterion. I recommend that the authors substantially expand the introduction to better explain how their work integrates with prior work on a quantum Chirikov criterion. I also recommend the authors include a brief introduction to past work towards a more general quantum KAM theorem to provide background as to how their work might 'advance understanding of a KAM threshold in low-dimensional cold quantum gases' (line 234).

***We have made revisions to the introduction, Section 2, and to Appendix A.3 to clarify our work in relation to past literature on the quantum Chirikov criterion and to make the connection to possible quantum extensions to the KAM theorem.

  1. It is unclear to me how the quantum post-selection on resonances is performed in Figs. 2 and 3. Does $(m_{max})_{min}$ set a lower bound on the perturbed eigenstate overlap with the unperturbed eigenstate denoting a resonance, as depicted in Fig. 3(a)? Or, does $(m_{max})_{min}$ set a lower bound on the participation number of the perturbed eigenstates, as depicted in Fig. 2? Or, does $(m_{max})_{min}$ set a lower bound on the number of unperturbed eigenstates within the energy window of the perturbation, as depicted in Fig. 1? If the latter, why is a non-integer value of $(m_{max})_{min} = 0.5$ chosen? I recommend further clarification in the text.

***This question is easy to answer: limiting the allowed variation of the state index $m$ to a unit-length $[-1/2,\,+1/2]$ interval excludes the classical resonances that span less than one quantum state.

  1. I attempted to derive the equation after line 169 using the process described, but was unsuccessful without additionally assuming that $m \sim m'$. Could the authors clarify the derivation?

***Indeed, we were also assuming $m \sim m'$. A corresponding correction was added.

  1. Eq. (15) is presented without explanation as to its derivation. Am I correct in inferring that it results from Eq. (13)? I recommend that a more detailed explanation of the derivation of Eq. (15) be included in the text.

***We have added the steps of the derivation and re-expressed Eq. (15) in a more transparent form.}

  1. I recommend that the transformation of coordinates required to derive Eq. (10) be described in the text with reference to appropriate figures. This should include an in-text definition of $\theta$.

*** It was not really a transformation, but rather a straightforward reinterpretation of the correction to the unperturbed Hamiltonian. We added subsections to the Appendix A so that the relevant section is easier to reference and find.

  1. Does the 'integrable reference system' mentioned on line 209 allow exact diagonalization of the perturbed system too? If not, how are the perturbed eigenstates determined?

***After the perturbation is added, the unperturbed is no longer integrable. The latter was diagonalized numerically exactly, using MatLab. The integrable reference system provides the basis in which the perturbation is represented for exact diagonalization.

  1. The label for Fig. 2(c) identifies the perturbation strength as $0.06$, while line 212 says $0.6$.

Thank you for catching this mistake. The correct strength is 0.06.

  1. I recommend that the term $M_{0}$ in Eq. (13) be defined in text and not just in the appendix.

***Done.

  1. Ref. [18] notes that the classical billiard system studied in this article is nonergodic for $\epsilon = \cos(\alpha)$ such that $\alpha$ is a rational or weak irrational multiple of $\pi$. I think that this should be mentioned in the article to contrast with the classical Chirikov prediction for arbitrarily small alpha (lines 311-312). Does this affect the statement on line 198?

***This is an excellent point. In fact, the observation in Ref. [18] defies intuition. Rational (within the allowed segment of the phase space) and weak irrational right triangular billiards show ergodicity, while the generic ones lack it. We have added a more detailed comparison of these results to Ref. [18], and included the conjecture that the Chirikov criterion is not able to detect the weak deviations from ergodicity described in [18].

  1. I infer that Eq. (11) is derived from lines 308-309 in the appendix. I recommend that this section of the appendix be referenced in text near Eq. (11).

***A long explanation added.

  1. What assumptions are made to produce the quantum Hamiltonian (3)? Is this Hamiltonian directly realised from Eq. (2) with the addition of the delta term an ansatz that is later validated by the Hilbert space truncation described on line 156?

***Equation 3 is not an Ansatz at all: it naturally emerges when the Hilbert space is restricted to the resonance line, in the semiclassical limit $n_{1,2} \gg 1$. The text around Eq. (3) has been written to address these questions, as well as to address these questions.

  1. According to Ref. [24], the eigenstate thermalization hypothesis posits that thermalization of observables occurs when the eigenstate expectation values of observables are smooth functions of energy. It is not immediately clear to me how the delocalization of a perturbed eigenstate in the basis of unperturbed eigenstates, as seen in Fig. 3(d), indicates thermalization of the observables $\hat{p}_{1}$ and $\hat{p}_{2}$. Could the authors please clarify their statement on lines 219-222?

***We have added several sentences to clarify.

  1. In Figs. 2-3 the resonance lines are hard to distinguish in grayscale, are quite thin, and overlap considerably. In particular, the black resonance lines in Fig. 2(c) cannot be distinguished from the PR background plot. Perhaps some bars denoting the endpoints of the resonance lines would be helpful for distinguishing overlapping resonances. The yellow resonance lines are particularly hard to distinguish against a white background.

***We've spent a significant amount of time optimizing the visuals. We were only able to implement a subset of suggestions by the Referee 1. However, we did add the bars and recognize the improvement.

  1. I recommend that the definition $\bar{n}=k/\sqrt{p^2+q^2}$ be included where $\bar{n}$ is defined.

***Corrected. In fact, its definition had to be included much earlier in the manuscript.

  1. I recommend that the paragraph starting at line 129 should reference appropriate figures to aid interpretability.

***Done.

  1. I recommend that the IPR be defined in text and described as a measure of purity, with appropriate bounds.

***We were not able to identify a placement of this definition that would be better than the one we had. We did correct a small mistake in the caption so the definition presented there should be more clear.

  1. I think the equations on lines 59 and 111 should be placed in LaTeX equation environments to aid readability.

***Done.

  1. I think that specific appendix sections should be referenced on lines 100, 177, 192, 195 and 200, and in footnote 2 on page 6.

***We introduced new subsections in the Appendix. Everything is properly referenced now.

  1. In Ref. [13], 'Kolovsky' is misspelled.

***Corrected.

  1. I think that all the equations in LaTeX equation environments should be numbered.

***This does not seem to be the policy of the journal. We understand the request but have decided to only number significant or referenced equations.

%%%%%%%%%%%%%%%%%%%%%%%%%% Referee 2

In the submitted paper by Yampolsky and others, the authors consider a very simple system of two quantum particles confined in a one-dimensional box as a workhorse to study the emergence of chaoticity induced by increasing mass imbalance between particles. To make the analysis as complete as possible, first, the classical system having two degrees of freedom is studied and its chaoticity is classified in terms of Chirikov resonances. The criterion is then naturally (but in my opinion not straightforwardly) generalized to the quantum case and implemented to the system under consideration.

The paper nicely exposes a relation between classical and quantum chaos appearing in this very simple but interesting system. The manuscript can be viewed also as a very good pedagogical example of treating classical-quantum many-body chaos puzzle with mathematical apparatus going far beyond simple level statistics. Personally, I find the paper quite well written and extremlly interesting -- probably deserving publication. However, before the final decision, I would like to ask the authors to consider the following remarks.

  1. At the end of Sec. 2 the authors describe quite shortly how to generalize the Chirikov criterion to the quantum scenario. This part is not very extended and contains sentences that are not fully justified. For example, ... analogous Chirikov structure is preserved...'',... quantum post selection ... eligible to enter the Chirikov criterion...'', etc. Are these facts well-known in the literature (then please add citations) or this is only a speculation of the authors (then it requires some more description to make argumentation more convincing). I would recommend rewriting this part to be more understandable.

***We have completely rewritten the end of Section 2 (also adding relevant references) to try and clarify what has been previously known about the quantum Chirikov criterion, and what is specific to the system under study in this article.

  1. In Sec. 3 the authors start a whole analysis by considering only antisymmetric (fermionic) states of two particles. Then they use them to calculate perturbative corrections related to different masses of particles. It is not clear why for fundamentally distinguishable particles the authors use only antisymmetric states rather than simple products of both wavefunctions. Can all two-particle states be included in the analysis and how it would change the results?

***Because we are considering hard-core interactions between different-mass particles, the (distinguishable) particles remain fixed in their order, i.e. $x_2 > x_1$. So we are only interested in half of the full configuration space and wave functions must vanish on the line $x_1 = x_2$. Antisymmetric states form a complete basis for the Hilbert space of functions built on this configuration space.

%%%%%%%%%%%%%%%%%%%%%%%%%% Referee 3

This article uses the so-called Chirikov's criterion for the onset of chaos in a very simple non-integrable quantum system. The results are rather remarkable, in the sense that the actual (change of level statistics) and Chirikov's estimates for the onset of chaos are in rather good agreement. I think the paper has great pedagogical value, and I have learnt myself a good deal of new physics reviewing the article. Derivations are sufficiently detailed as well, and while semi-rigorous, I would think its not out of reach for most researchers in theoretical physics. It is probably worth publication since it may trigger new research in more relevant systems (like non-integrable many-body systems) and allow for estimations of the onset of chaos in these settings.

I think the authors have cut down the discussion on the classical Chirikov's criterion a little too much. Perhaps my ignorance is fooling me here, but my guess is that this criterion is not well known to most researchers in quantum physics. It certainly wasn't known to me. I would suggest the authors use a bit more space to introduce the classical version. However, its connection to KAM theory seems quite OK to me as it stands.

I would also like the authors to try and motivate their choice of system, besides method-wise, a little bit more. I mean, some physical motivation could be interesting and would be very welcome. Even if not for this particular case, I can think of a system that can be discussed in passing even in the conclusions: two particles (of equal masses or otherwise) with an impurity (a fixed potential) at the origin. This system is highly non-trivial and, at least on a lattice, supports interesting physics such as bound states in the continuum (see Phys. Rev. Lett. 109, 116405 ).

A minor comment: I understand "verbatim" is used somewhat extensively in the math literature. It's however uncommon in physics. Could the authors use "one-to-one" instead?

Other than the above comments, I think this is a very neat paper, and I am certain everyone will enjoy reading it and learning it as much as I have.

Requested changes:

  1. Add some further discussion about the Chirikov criterion in Classical Mechanics.

***We added an explanatory introductory paragraph to the Chirikov...'' subsection. Also, to enhance readability of the wholeClassical Analysis...'' Appendix, we introduced a thorough sectioning there, divided the principal figure onto subfigures, and after that, inserted several key references to the sub-sections and sub-figures, throughout the text.

  1. Add physical motivation for the system studied.

Regretfully, we were not able to find a better motivation than the one given in the second paragraph of the introduction. However, in the conclusion, we have expanded our discussion of the relation of this toy model to quantum thermalization as a motivating application.

  1. Change "verbatim" to non-latin English.

***Corrected.

---

## Editorial Decision

published